# ENCODER-DECODER NETWORK AS SEQUENCE LOSS FUNCTION FOR SUMMARIZATION

## ABSTRACT

We present a new approach to defining a sequence loss function to train a summarizer by using a secondary encoder-decoder as a loss function, alleviating a shortcoming of word level training for sequence outputs. The technique is based on the intuition that if a summary is a good one, it should contain the most essential information from the original article, and therefore should itself be a good input sequence, in lieu of the original, from which a summary can be generated. We present experimental results where we apply this additional loss function to a general abstractive summarizer on a news summarization dataset. The result is an improvement in the ROUGE metric and an especially large improvement in human evaluations, suggesting enhanced performance that is competitive with specialized state-of-the-art models.

## 1  INTRODUCTION

Neural networks are a popular solution to the problem of text summarization, the task of taking as input a piece of natural language text, such as a paragraph or a news article, and generating a more succinct text that captures the most essential information from the original.

One popular type of neural network that has achieved state of the art results is an attentional encoder-decoder neural network See et al. (2017); Paulus et al. (2018); Celikyilmaz et al. (2018). In an encoder-decoder network, the encoder scans over the input sequence by ingesting one word token at a time to create an internal representation. The decoder is trained to compute a probability distribution over next words conditioned on a sequence prefix. A beam search decoder is typically used to find a high likelihood output sequence based on these conditional word probability distributions.

Since the next word depends heavily on previous words, the decoder has little hope of a correct distribution over next words unless it has the correct previous words. Thus the decoder is typically trained using *teacher forcing* Williams & Zipser (1989), where the reference sequence prefix is always given to the decoder at each decoding step. In other words, regardless of what distributions are output by the decoder in training for timesteps $(1, ..., t-1)$, at timestep $t$, it is given the reference sequence prefix $(y_1^*, ..., y_{t-1}^*)$ and asked to output the distribution $P(y_t | y_1^*, ..., y_{t-1}^*)$.

Teacher forcing suffers from *exposure bias* Ranzato et al. (2016). At test time, the input to the decoder is not the reference sequence prefix $(y_1^*, ..., y_{t-1}^*)$ but a $(y_1, ..., y_{t-1})$ constructed based on the decoder's own outputs. Since the decoder is not perfect, as $t$ increases so too does the disparity between these two sequence prefixes, causing an increasing mismatch between training and test conditions.

Training at the sequence level can alleviate this discrepancy, but requires a differentiable loss function. In the Related Work section we review previous efforts.

We present a novel approach to address the problem by defining a loss function at the sequence level using an encoder-decoder network as a loss function. In training, the summarizer's beam search decoded output sequence is fed as input into another network called the *recoder*. The recoder is an independent attentional encoder-decoder trained to produce the reference summary.

Our experiments show that adding the recoder as a loss function improves a general abstractive summarizer on the popular CNN/DailyMail dataset Hermann et al. (2015); Nallapati et al. (2016),

achieving significant improvements in the ROUGE metric and an especially large improvement in human evaluations.

## 2    ATTENTIONAL ENCODER-DECODER MODEL FOR SUMMARIZATION

We first give a high level overview of an attentional encoder-decoder model for sequence to sequence learning. We present the specific abstractive model of See et al. (2017), which serves as the baseline model in our experiments for comparison. We chose this as the baseline model because it is a general and popular model whose concepts have often appeared in other abstractive summarization works, and its results have often been used as a comparison baseline Chen & Bansal (2018); Li et al. (2018); Celikyilmaz et al. (2018).

The ideas presented in this paper are largely independent of the specific summarizer and may be applicable to the training of other sequence to sequence models, many of which are also trained at the word loss level using teacher forcing. Since our focus is to demonstrate the effectiveness of our new loss function, the generality of the See et al. (2017) model, which does not account for specifics of the summarization problem such as article or sentence structure Li et al. (2018), suits our purpose well.

### 2.1    INPUTS AND OUTPUTS

The source article and target summary are each treated as a sequence of tokens. A token represents either a word or a punctuation mark. Let $(x_i)$ be the sequence of tokens in an article or input sequence, where $i$ ranges from 1 to the length of the input sequence. Let $(y_t)$ be the sequence of word tokens in the summary or output sequence, where $1 \leq t$.

The goal is to train a neural network to compute, at each timestep $t$, the probability distribution $P(y_t|y_1, ..., y_{t-1})$ over next tokens $y_t$ given previous tokens $(y_1, ..., y_{t-1})$. In the teacher forcing training method, the network is trained to minimize the loss $J_{\mathrm{ml}}$, also called the *maximum likelihood* loss, defined as the average over timesteps $t$ of $(J_{\mathrm{ml}})_t = -\log P(y_t^*|y_1^*, ..., y_{t-1}^*)$ where the sequence prefix $y_1^*, ..., y_{t-1}^*$ is from the reference summary.

At test time, the network computes $P(y_t|y_1, ..., y_{t-1})$ where $y_1, ..., y_{t-1}$ are chosen by a beam search decoder based on previously output distributions. The sequence ends when the special token STOP, appended to the end of all reference summaries in training, is output.

### 2.2    ENCODING

Each $x_i$ is embedded as a dense vector $w_i$ using a $v \times d$ weight matrix $W_{\mathrm{emb}}$, where $v$ is the size of the vocabulary and $d$ is the number of hidden dimensions, a hyperparameter. The weights $W_{\mathrm{emb}}$ are learned as part of training. Each $w_i$ is given in order as input into an LSTM to get hidden states $(h_i^f)$, and also in reverse order into another LSTM to get backward states $(h_i^b)$. The two state sequences are then stacked to form hidden states $(h_i)$ for reference by the decoding phase's attention mechanism.

### 2.3    DECODING WITH ATTENTION

The decoding phase also uses an LSTM. At each output timestep $t$, a token $y$ is embedded as a dense vector using the same $W_{\mathrm{emb}}$ and fed into the LSTM, and the LSTM outputs a probability distribution over next tokens. For $t = 0$, $y$ is the special placeholder symbol START. For $t > 0$, when training using teacher forcing, $y = y_{t-1}^*$ is the corresponding token from the reference sequence. When beam search decoding at test time, $y = y_{t-1}$ is chosen based on the decoder's output probabilities at step $t - 1$. This difference in information about previous output tokens causes discrepancy between training and test conditions.

The decoder makes use of an additional piece of information called *attention* Bahdanau et al. (2015), which allows it to "read" the encoder's output states. At each timestep with decoder LSTM state $s_t$, the attention vector is computed as

$$a_t = \mathrm{softmax}(g^T \tanh(W_h h_i + W_s s_t + b_{\mathrm{attn}}))$$

where $g$, $W_h$, $W_s$, and $b_{\text{attn}}$ are parameters to be trained. This vector gives a distribution over encoder states at timestep $t$. The attentional *context vector* $h_t^*$, a "reading" of encoder states, is then computed as

$$h_t^* = \sum_i (a_t)_i h_i$$

This context vector is concatenated with the LSTM state $s_t$ to produce the dense vector outputs $q_t = V[s_t, h_t^*] + b$, where $[]$ denotes concatenation of vectors. Finally, another linear transformation using a weight matrix $V'$ and weight vector $b'$ is used to map these outputs in the hidden space, through a softmax layer, into output probabilities over the vocabulary. That is,

$$P_{\text{vocab}}(y_t|y_1, ..., y_{t-1}) = \text{softmax}(V'q_t + b')$$

where $V$, $V'$, $b$, $b'$ are trainable parameters.

## 2.4 POINTER GENERATOR AND COVERAGE

The pointer generator mechanism allows the decoder to copy a token directly from the input. It is particularly helpful for out-of-vocabulary tokens. The decoder "copies" source tokens with a distribution based on the attention vector $a_t$ and mixes it in with weight $1 - p_{\text{gen}}$, where $p_{\text{gen}}$ is a trainable function of the context vectors, states, and next word input. The final distribution $P(y_t|y_1, ..., y_{t-1})$ is the combined

$$p_{\text{gen}} P_{\text{vocab}}(y_t|y_1, ..., y_{t-1}) + (1 - p_{\text{gen}}) \sum_{i:x_i=y_t} a_i^t$$

The summarizer loss function $J^{\text{S}} = J_{\text{ml}}$ is computed based on this distribution.

In one model variant, a *coverage* loss $J_{\text{cov}}$ that encourages attention vectors $a_t$ to look at different encoder states $i$ is also added to the final loss function. That is, $J^{\text{S}} = J_{\text{ml}} + J_{\text{cov}}$.

Full details on the pointer-generator and coverage mechanisms are found in See et al. (2017). We have omitted them here for brevity. The recoder does not depend on these particulars of the summarization model. We focus on the quantities $P_{\text{vocab}}$, $P$, and the total summarizer loss $J^{\text{S}}$. Specific details of how these are computed are not needed in the following sections.

## 3 RECODER

We are now ready to present our main contribution. The recoder is a neural network model that takes the decoded sequence from the summarizer as input and is trained to output the reference summary. The purpose of the recoder is to serve as a sophisticated loss function to help train the summarizer, backpropagating errors to the summarizer's decoded output sequence during training. The intuition is that a good output sequence from the summarizer should allow the recoder to produce a good summary.

In principle, the recoder can be any sequence to sequence model that can backpropagate to its inputs. One obvious choice is to use the same model structure as the summarizer itself. For our experiments we found it was sufficient to use the same network structure with lower dimensions, with the same attentional pointer-generator encoder-decoder network as the summarizer, but with half the number of hidden dimensions (from 256 to 128) and a GRU Cho et al. (2014) instead of an LSTM in the encoder. This helped reduce the amount of memory required in training.

### 3.1 RECODER INPUTS

Our first task is to represent the summarizer's decoded outputs as inputs to the recoder in a way that is differentiable. A beam search decoder will find a sequence $(y_1, y_2, ...)$ of high average log probability $-\log P(y_t|y_1, ..., y_{t-1})$ over timesteps $t$. We cannot use this discrete token sequence directly, but we can look at the underlying signals that determine the choice of each token.

The output token $y_t$ is chosen based on the computed probability distributions $P_t(y_t|y_1, ..., y_{t-1})$. Let us denote this distribution by $P_t$ for short. For a beam search of width $k$, the chosen token $y_t$ will have a probability in $P_t$ that is among the $k$ highest. The exact choice is determined by the beam

search mechanism, so we do not have a continuous function that relates $P_t$ directly to $y_t$. However, $P_t$ does determine the range of choices, and feeding it as an input to the recoder can ensure that it contains the right information. Since each $P_t$ is computed by the summarizer based on the decoded prefix sequence $(y_1, ..., y_{t-1})$, propagating errors to $P_t$ improves the summarizer via exposure to what it would see at test time, even if the summarizer is not "aware" of the search mechanism and cannot optimize for it.

Since $P_t$ has dimensions equal to the size of the vocabulary ($v = 50000$ in the experiments), we multiply $P_t$ by a weight matrix to produce a dense representation in a lower dimensional space (128 in the experiments), just as we did for the one-hot token inputs of the summarizer. While we can use a new weight matrix here, reusing the same embedding matrix $W_{\mathrm{emb}}$ as the summarizer helps avoid increasing the number of parameters. For the same reason we also reuse the summarizer's mapping $V'$ back to vocabulary space. Together they account for 90% of the summarizer's parameters.

The input to the recoder is the sequence $(w_t^R)$ where

$$w_t^R = P_t W_{\mathrm{emb}}$$

with $P_t$ treated as a vector of length $v$. By training the recoder to output the reference summary, errors are propagated to its inputs $w_t^R$ and then to $P_t$.

Note that although the summarizer's input $w_i$ is in effect $W_{\mathrm{emb}}$ multiplied with one-hot vectors corresponding to $x_i$, $P_t$ is the output of a softmax function and will be more evenly distributed than a one-hot vector. Embedded representations $w_t^R$ will thus be a weighted sum of embedding vectors instead of individual ones like $w_i$. The recoder can accommodate this difference since it is an independent network with its own parameters.

## 3.2 Recoder Training

The recoder can be trained jointly with the summarizer from scratch. We have found that it also suffices to add the recoder to the pretrained summarizer and continue training jointly using their combined losses.

The recoder is trained using teacher forcing to produce the reference summary, analogously to $J_{\mathrm{ml}}$. To be specific, we minimize the loss function $J_{\mathrm{ml}}^{\mathrm{R}}$ equal to the average of $-\log P^{\mathrm{R}}(y_t^* | y_1^*, ..., y_{t-1}^*)$ across timesteps $t$, where $P^{\mathrm{R}}$ denotes the recoder's output distributions. This maximum likelihood loss is sufficient for the recoder because its output is never used for decoding at test time.

## 3.3 Length Loss

Meaningful recoder output depends on relevant information from the original article being encoded in its input $w_t^R$. However, $J_{\mathrm{ml}}^{\mathrm{R}}$ only places requirements on the presence of information. We have not placed any requirements on brevity, so longer sequences $w_t^{\mathrm{R}}$ would likely yield better results under this loss metric, barring any confounding effects from too much information. The effect is only constrained because recoder training is performed jointly with the summarizer, and summaries that are longer than reference summaries would tend to do worse with respect to $J^{\mathrm{s}}$. Intuitively, $J_{\mathrm{ml}}^{\mathrm{R}}$ encourages recall of information. We can add a loss function on length to counterbalance for precision.

The end of an output sequence is determined when the special STOP token is output. Actual words are likely to convey more useful information to the recoder, so training using $J_{\mathrm{ml}}^{\mathrm{R}}$ lowers the STOP token's output probability. We can control length by applying a loss to every other token for timesteps beyond a desired length. We define the *length loss* as the average over $t$ of

$$(J_{\mathrm{len}})_t = \lambda \cdot \mathtt{Penalty}(t)(1 - P_t(\mathrm{STOP}))$$

The function $\mathtt{Penalty}(t) \geq 1$ defines a penalty curve for timesteps beyond a desired length, while hyperparameter $\lambda$ defines the desired tradeoff between precision and recall.

Other ways to control output length are possible, such as by explicitly adding the desired length into decoder state Kikuchi et al. (2016); Fan et al. (2017), although these methods require changes to the model.

| | |
|---|---|
| **Article** | some teenagers get driving lessons from their parents . other teens are taught by licensed instructors . but malia obama is n't your average 16-year-old : her driving lessons were provided by the u.s. secret service . asked who taught malia how to drive , first lady michelle obama told celebrity chef and daytime talk-show host rachael ray in an interview that it was the armed agents who provide around-the-clock security for the family . [...] |
| **pgen_cov** | first lady michelle obama told celebrity chef and daytime talk-show host rachael ray in an interview that it was the armed agents who provide around-the-clock security for the family . mrs. obama has n't driven herself in seven or eight years . |
| **pgen_cov+recoder** | **malia** obama , seen with her mother michelle obama in april 2009 , reportedly was taught how to drive by secret service agents . but **malia** obama is n't your average 16-year-old : her driving lessons were provided by the u.s. secret service . asked who taught **malia** how to drive , first lady michelle obama said in an interview that it was the armed agents who provide around-the-clock security for the family . |
| **reference** | michelle obama told talk-show host rachael ray that secret service agents taught her daughter **malia** how to drive . mrs. obama has n't driven herself in seven or eight years , she said . she added that driving gives **malia** ' a sense of normalcy , ' helping her feel like the rest of her friends who are also driving . |

Figure 1: Sample output from the CNN/DailyMail dataset.

## 3.4 COMBINED LOSS

The final recoder loss $J^{\mathrm{R}}$ is comprised of the teacher forced recoder loss and the length loss

$$J^{\mathrm{R}} = J^{\mathrm{R}}_{\mathrm{ml}} + J_{\mathrm{len}}$$

Abstracting away individual components of the summarizer's and recoder's loss functions, the end to end training is performed using a combination of their losses $J$:

$$J = J^{\mathrm{S}} + J^{\mathrm{R}}$$

## 3.5 EXAMPLE

In Figure 1 comparing output summaries from the baseline `pgen_cov` model trained with $J^{\mathrm{S}}$ (described in Experiments section below) and the `pgen_cov+recoder` model additionally trained with loss $J^{\mathrm{S}} + J^{\mathrm{R}}$, the latter's output contains mention of "malia", a relevant name in the article that also appears in the reference summary.

If we trained only with the recoder loss $J^{\mathrm{R}}$, we would encourage the summarizer to output the right information, but the output may not conform to a language model, except to the extent it helps make the output more intelligible for the recoder. For the purpose of illustration, we continued training the `pgen_cov` model using only $J^{\mathrm{R}}$. It produced the following summary for one of the articles in the CNN/DailyMail dataset:

```
boston 's miserable winter is now also
its snowiest season it had a special
2.9 inches pushed the city into 108.6
inches in one season .
```

This example output contains some relevant information but has grammatical errors.

## 4 RELATED WORK

Summarization is a well studied problem, including both extractive Zhou et al. (2018) and abstractive approaches Zeng et al. (2016); Nallapati et al. (2016); Rush et al. (2015). Many of the existing abstractive models are based on encoder-decoders and trained using teacher forcing. In this work we focus on improving the training of models such as these, and we have picked one particular model as the baseline for improvement. Many aspects of the See et al. (2017) baseline model also appear in other works. For example, Li et al. (2018) applies a similar attentional coverage mechanism at the sentence level, while Chen & Bansal (2018); Celikyilmaz et al. (2018) employ a pointer-generator mechanism at the word or sentence levels.

While more recent work, such as Celikyilmaz et al. (2018); Chen & Bansal (2018); Li et al. (2018), have reported better ROUGE results on the CNN/DailyMail dataset, we chose the See et al. (2017) model for its generality. By using an unspecialized model with commonly occurring elements as baseline, we hope the same concepts can apply to more advanced and specialized models.

## 4.1 REINFORCEMENT LEARNING

There are a number of related techniques to address exposure bias from teacher forcing in a sequence learning problem. One class of such techniques is motivated by the ideas of *reinforcement learning (RL)* for training *actors* to make discrete decisions Paulus et al. (2018); Ranzato et al. (2016); Rennie et al. (2017); Bahdanau et al. (2016).

The encoder-decoder is analogous to an actor being trained to learn a policy for choosing an output word at each time step. These decisions are evaluated in training not based on its output probabilities, but based on a *reward* function defined on the full output sequence, selected via beam search or other methods.

Broadly speaking, there are a couple ways to turn rewards into gradients for training probability distributions at each timestep. One technique is the REINFORCE algorithm, Williams (1992); Rennie et al. (2017); Paulus et al. (2018), which is based on computing expected future rewards, and the related technique of *minimum risk training* in Makino et al. (2019). Another approach is to train an estimator called a *critic* model to estimate expected rewards, as in Ranzato et al. (2016); Bahdanau et al. (2016).

The common key to these approaches is the reward function. Typically the reward is defined based on n-gram overlap, such as the use of ROUGE Paulus et al. (2018); Rennie et al. (2017), BLEU Ranzato et al. (2016), or METEOR Bahdanau et al. (2016) scores. The reward function serves as one component, while a technique such as REINFORCE or a critic model serves as the second component. Together they turn full sequences into gradients for training the output of each timestep.

Cast in these terms, the recoder serves the role of both components. However, instead of training toward a heuristic such as ROUGE, the recoder uses an encoder-decoder network, allowing it to account for complexities in evaluation of quality that are as sophisticated as the model being trained. An algorithm can only be as good as the metric toward which it is trained, and the recoder helps ease this upper bound on quality. In short, the key difference from reinforcement learning approaches is that the recoder replaces a simple reward heuristic with an encoder-decoder and its loss function.

## 4.2 SEQUENCE OUTPUT LOSS

Another approach to account for the beam search decoded output sequence is Wiseman & Rush (2016). In this work, beam search is performed as part of training, and a loss is defined that penalizes the reference sequence prefix falling out of the beam. In Goyal et al. (2017) on two non-summarization word sequence tasks, a differentiable approximation to beam search is used, and loss is defined on the sequence using Hamming distance. While these approaches can account for the decoder test time sequence output, they do not have the flexibility to credit alternative summaries that differ slightly in phrasing. In comparison, our approach can account for a range of correct summaries that differ from one another, regardless of subsequence overlap. Since the recoder is itself a type of summarization network, it too can keep pace should further improvements in summarizers be developed that output more varied summaries of high quality. However, using a differentiable approximation to beam search enhances backpropagation, and may be a direction for future improvement.

The idea of re-encoding an output summary also appears in Chu & Liu (2019), where Straight-Through Gumbel-Softmax reparameterization Jang et al. (2017) is used to derive a loss function that allows backpropagation. Their `reconstruction cycle loss` variant is closest in concept to our work, except that since there is no reference summary in their problem, they train their analogue of the recoder to produce the original source. We did not take this approach because in general the summary is a lossy representation of the original, so training the recoder to produce it would subject it a large source of error that it cannot possibly reduce.

In the extension Xu et al. (2019) to *scheduled sampling* for machine translation, a decoded sequence prefix is used in training to predict the next reference word after weighting over possible alignments, which allows for flexibility in word ordering. For the summarization problem, alternative summaries that differ by more than word alignment can still be of high quality.

### 4.3 Complementary Models

The idea of using a second model to help train the primary model has appeared in other contexts. In Sennrich et al. (2016); He et al. (2016) for machine translation, the decoded output of a translation model is fed as input into a second *backtranslation* model. The two separate models translate back and forth between source and target languages. We can think of the backtranslation model from target to source language as serving an analogous role to the recoder, although the aim of these works has been to generate synthetic training data to augment limited human training data, rather than to compute gradients directly in an end-to-end model for the purpose of improving the original network. In contrast to the translation problem, we also cannot expect the original article to be recreated from the summary, which in general will be a lossy representation of the article, so the recoder is asked to recreate the reference summary and not the original article. An analogous idea also appears in the problem of sentence compression Miao & Blunsom (2016); Févry & Phang (2018), where decoding from a shortened version of a sentence back to the original sentence is analogous to backtranslating from target to source language.

In the related work Xia et al. (2018) on dual learning, the two models for the forward and backward problems share parameters to mutual benefit. In contrast, the recoder shares no parameters with the summarizer except vocabulary mappings, relying instead on direct backpropagation.

In Kim & Rush (2016), the decoded output sequence of a translation model is used to train a *student* model to mimic the original model's output. The aim there is to better distill knowledge into a smaller student model, not to improve the original model.

## 5 Experiments

We ran experiments to test the effectiveness of adding the loss function to the attentional encoder-decoder summarization model of See et al. (2017). The Tensorflow code, which is based on their published code, will also be made public along with trained models.

We started with the provided trained summarizer models and added on recoder losses and continued training. The additional training consumes about double the memory and computation time, as it requires a beam search decoding as part of training, using a beam width of 4. The additional time was about 24 hours for each model, using an Nvidia Tesla P100 GPU on a cloud service provider. Test times are not affected since the recoder is only used during training.

We ran experiments using the two versions of the model from See et al. (2017), with and without the coverage mechanism, as baselines. The model with coverage is `pgen_cov`, while `pgen` omits the coverage mechanism. In terms of loss functions, `pgen_cov` is trained using a loss of $J^S = J_{\mathrm{ml}} + J_{\mathrm{cov}}$ as presented above, while `pgen` is trained using $J^S = J_{\mathrm{ml}}$. In either case, the comparison models with recoder is additionally trained using $J^S + J^R$.

We used the summarizer's $(P_{\mathrm{vocab}})_t$ directly in place of $P_t$ in computing recoder inputs, bypassing backpropagation to the pointer-generator mechanism, which is most helpful for out-of-vocabulary words that will be treated as `UNK` for $w_t^R$ anyway. This yielded similar results while converging faster and requiring less computational resources.

We fixed the length penalty `Penalty`$(t)$ to be a graduated curve $1.04^{\max(t-0.8L,0)}$ where $L$ is the length of the reference summary for the example. We experimented with different settings of $\lambda$ to see its effects on output lengths and quality. The final setting of $\lambda = 0.1$ used for the ROUGE comparison in Table 1, discussed below, was selected based on its having the highest ROUGE-1 score on the validation set, where scores appeared in the same relative order as on the test set results shown in Table 2.

### 5.1 ROUGE Comparisons

In line with the baseline and for comparison with previous work, we first assess quality using the ROUGE metric Lin (2004). The recoder can in principle capture more complex notions of quality than ROUGE, so this heuristic cannot fully capture all aspects of improvement that the human evaluations described below can. Nonetheless, since we do not train toward this metric, it should serve as an independent indicator of overall quality improvements.

| Model | ROUGE-1 | ROUGE-2 | ROUGE-L |
|---|---|---|---|
| pgen (baseline) | 36.08 | 15.60 | 32.95 |
| **pgen+recoder** ($\lambda = 0.1$) | **37.07** | **16.02** | **33.83** |
| pgen_cov (baseline) | 39.47 | 17.37 | 36.26 |
| **pgen_cov+recoder** ($\lambda = 0.1$) | **40.44** | **18.15** | **36.90** |
| lead-3 | 39.94 | 17.45 | 36.23 |
| hierarchical Li et al. (2018) | 40.30 | 18.02 | **37.36** |
| pgen (postproc) | 39.28 | 17.40 | 36.13 |
| pgen+recoder (postproc) | 39.42 | 17.30 | 36.18 |
| pgen_cov (postproc) | 39.83 | 17.56 | 36.61 |
| pgen_cov+recoder (postproc) | 40.71 | 18.20 | 37.12 |

Table 1: ROUGE F1 scores for 1-gram, 2-gram, and longest common subsequence on CNN/DailyMail test set. The scores shown for `pgen` and `pgen_cov` differ slightly from that listed in See et al. (2017), e.g. 39.47 vs 39.53 for ROUGE-1. The pretrained models provided with their published code resulted in lower scores (differences may be due to changes in the underlying software libraries over time), but we were able to achieve these comparable scores by further training with a lower learning rate.

The results are shown in Table 1. For `pgen`, we see that adding the recoder loss results in an improvement in scores from (36.08, 15.60, 32.95) to (37.07, 16.02, 33.83). For the higher scoring `pgen_cov` baseline, adding the recoder loss improves scores from (39.47, 17.37, 36.26) to (40.44, 18.15, 36.90). In both cases we see close to a 1 point improvement in ROUGE-1 scores, and a smaller improvement in ROUGE-2 and ROUGE-L scores versus the baseline.

We see that adding the recoder to two variations of a strong abstractive summarizer has improved their performance by a significant amount, even though we have not changed the models, only the loss function used to train them. These scores are also an improvement over using the first 3 sentences from the article as the summary (lead-3), which had outperformed the baseline models.

To get a sense of the magnitude of improvement, we look at one recently published model that uses the same non-anonymized version of the dataset without further processing, allowing for a relatively direct comparison. The "hierarchical" model of Li et al. (2018) is an abstractive model that accounts for sentence level structure. Overall results are close to `pgen_cov+recoder`, which has higher ROUGE-1 and ROUGE-2 scores but lower ROUGE-L. Applying the recoder to a generic baseline has resulted in ROUGE gains that match that of a model with more advanced mechanisms.

There are other sophisticated abstractive models with higher reported ROUGE scores, especially those that are extractive hybrids Chen & Bansal (2018) or employ RL techniques Paulus et al. (2018); Chen & Bansal (2018); Celikyilmaz et al. (2018), with a high of 41.69 ROUGE-1 score in the latter work after eliminating duplicate trigrams in post-processing. If we also eliminated duplicate trigrams in post-processing, the `pgen_cov+recoder` scores would be (40.71, 18.20, 37.12), but a full treatment should also eliminate duplicate trigrams during training, requiring a far more complex implementation. These models are generally more tuned to the specific problem and metric, such as accounting for sentence boundaries or including the ROUGE metric as reward, and it is reasonable that they achieve higher scores on this metric.

Since the recoder has the potential to capture notions of similarity beyond n-gram overlap as used in ROUGE, we performed further evaluation using human readers.

| Parameter $\lambda$ | Length | R-1 | R-2 | R-L |
|---|---|---|---|---|
| 0 | 87.1 | 40.17 | **18.17** | 36.80 |
| 0.1 | 74.9 | **40.44** | 18.15 | **36.90** |
| 0.2 | 68.7 | 40.29 | 18.02 | 36.70 |
| 0.3 | 52.1 | 39.18 | 17.24 | 35.52 |
| pgen_cov (baseline) | 66.1 | 39.47 | 17.37 | 36.26 |

Table 2: Length and ROUGE scores on CNN/DailyMail test set for different settings of length loss weight $\lambda$ on the `pgen_cov+recoder` model.

| Model | Overall | Readability | Relevance |
|---|---|---|---|
| pgen_cov+recoder ($\lambda = 0.1$) | 181 (60.3%) | 164 (54.7%) | 188 (62.7%) |
| pgen_cov+recoder ($\lambda = 0.2$) | 165 (55.0%) | 160 (53.3%) | 166 (55.3%) |

Table 3: Human rater preference versus pgen_cov on 300 random examples from the CNN/DailyMail test set. 95% confidence intervals for pgen_cov+recoder ($\lambda = 0.1$) preference are (54.6%, 65.9%) overall, (48.8%, 60.4%) readability, and (56.9%, 68.2%) relevance, and (49.2%, 60.7%), (47.5%, 69.1%), (49.5%, 61.1%) respectively for pgen_cov+recoder ($\lambda = 0.2$).

## 5.2 HUMAN EVALUATIONS

We ran human evaluation experiments comparing both the pgen_cov+recoder $\lambda = 0.1$ model and the $\lambda = 0.2$ model against the pgen_cov baseline. The $\lambda = 0.2$ model's average summary length of 68.7 is close to the baseline's 66.1, which minimizes differences due to length.

From the CNN/DailyMail test set where the model gave beam search decoded outputs that differed from the baseline (95.2% for $\lambda = 0.1$ model and 95.5% for $\lambda = 0.2$ model, out of 11490 examples), we randomly sampled 300 examples from each model. Workers from Amazon Mechanical Turk were shown up to the first 400 tokens from the article (same as model inputs), the reference summary, and the two generated summaries, randomly assigned to side A and side B. They were asked to select an overall preference between A and B, and then preference in terms of readability and relevance. Since we do not need a confident score for any one example for the purpose of evaluating the model, we limited each example to one worker and each worker to 5 examples to increase diversity across example-worker pairs.

Workers were asked the following questions with no other guidance on interpretation:

- `Which summary is better overall?`
- `Which summary has better readability?`
- `Which summary contains information that is more relevant?`

The preference results are shown in Table 3. We see that pgen_cov+recoder was preferred overall 60.3% ($\lambda = 0.1$) and 55.0% ($\lambda = 0.2$) of the time over the pgen_cov baseline. If we account for the remaining 4.8% and 4.5% of cases where the models gave outputs identical to the baseline by assigning equal preference to them, the preference ratios would be adjusted slightly to 59.8%, 54.8% overall and 62.1%, 55.1% relevance respectively. The overall improvement may be largely explained by the improvement in relevance, as the relevance preference was different from overall preference for only 19 and 25 examples respectively.

The most direct comparison from previous work may be the human evaluations of Chen & Bansal (2018) that had also compared their model (`rnn-ext+abs+RL+rerank`), with a ROUGE-1 score of 40.88, against the pgen_cov baseline. They allowed a choice of "Equally good/bad" which our survey did not, but if we assign those ratings equally to the two sides, their results suggest a preference for relevance 52.8% of the time.

If we similarly split "equal" ratings in the head-to-head human evaluations in Celikyilmaz et al. (2018), their overall preference results would be 59.3%. However, their comparison pitted a model (`m7`) with a ROUGE-1 score of 41.69 against a baseline model (`m3`) that had only achieved a ROUGE-1 score of 38.01, which is lower than our pgen_cov comparison baseline.

Differences such as post-processing and survey format make these comparisons imprecise, but they give us a sense of the magnitude of improvement reflected by a 54.8%-59.8% overall preference over the pgen_cov baseline.

## 6 CONCLUSION

We have presented the use of an encoder-decoder as a sophisticated loss function for sequence outputs in the problem of summarization. The recoder allows us to define a differentiable loss function on the decoded output sequence during training. Experimental results using both ROUGE and human evaluations show that adding the recoder in training a general abstractive summarizer

significantly boosts its performance, without requiring any changes to the model itself. In future work we may explore whether the general concept of using a model as loss function has wider applicability to other problems.

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
