# OpenReview forum: "Encoder-decoder Network as Loss Function for Summarization"
_ICLR.cc/2020/Conference — Reject_

### Official Review · AnonReviewer2 · 2019-10-15
**Official Blind Review #2**

**Rating:** 1

**Review:**

This paper proposes to use an additional component to the commonly used encoder-decoder approach for summarization, which is referred to as the recoder, which is an RNN-syle component that takes the output of the decoder. The intuition offered in the paper is that a good summary should produce itself via the recoder network, and in training it together with the original encoder-decoder it should improve its performance as it would be able to capture more than what the word-level loss does.

I have the following objections to this paper:
- I can't see why this extra component should improve the quality of the summary produced. If say our encoder-decoder architecture models the training data perfectly, then a recoder that does not do anything would be the right choice. Taking this further, a recoder could actually be fixing problems in the output of the decoder, and thus not providing a good training signal. It doesn't make sense to my that a summary should produce itself via a neural network, unless we are training an auto-encoder. The experiments validate this; the difference in ROUGE score are less than a point, which is the kind of fluctuation one expects due to random seed differences, etc.

- I find it odd that ROUGE score is dismissed as a loss to train against (referred to as a heuristic in the end of section 4.1), but then 1 ROUGE point difference is considered a "significant" improvement (making such claims without statistical significance testing is misleading). Sure it has flaws, if there is something better why not use it for evaluation? Furthermore, claiming that the recoder does a better job requires evidence. Why not train this extra function against human judgements? Assuming that what is wanted is to train a model that estimates the quality of the summary, it would make sense to look at the approaches used for the similar task of machine translation quality estimation: https://www.mitpressjournals.org/doi/pdf/10.1162/tacl_a_00056

- The approach is in my view a kind of actor-critic approach; the recoder could play that role, and in fact the Bahdanau et al. 2016 paper cited does this. However no comparison is offered, be it theoretical or experimental. Furthermore, the criticism that sequence-level training requires differentiable losses is incorrect; MIXER for example does train against scores such as ROUGE that are not differentiable. Furthermore, the BSO approach by Wiseman and Rush (2016) cited does give a continuous output to optimize for seq2seq that is asked for in the beginning of page 4.

- In the experiments, how was the length penalty determined to be the graduated curve mentioned in section 5? I would expect to have comparison against other approaches that try to train encoder-decoder to improve summarization, such as those mentioned in section 4.1.

- On the human evaluation experiments: the difference between the two models is quite small, especially given that the workers were not allowed to say that the models were equally good/bad. Furthermore, there is no inter-annotator agreement, as each comparison was done by a single crowd worker. Showing only the first 400 tokens of the original document would incorrectly disadvantage models selecting content from later in the articles. Finally, showing the reference summary creates another bias, since equally good summaries can disagree on what content to include. It might be helpful to look at some recent work on manual evaluation of summarization that tried to address these issues: https://arxiv.org/abs/1906.01361



**Experience Assessment:**

I have published one or two papers in this area.

**Review Assessment: Checking Correctness Of Derivations And Theory:**

N/A

**Review Assessment: Checking Correctness Of Experiments:**

I carefully checked the experiments.

**Review Assessment: Thoroughness In Paper Reading:**

N/A

---

> ### Author Response · Authors · 2019-11-05
> **You have misunderstood the paper entirely.**
>
> You appear to have misunderstood the recoder and how it helps the summarizer, and this is causing confusion for the questions below.
>
>
> > The intuition offered in the paper is that a good summary should produce itself via the recoder network.... It doesn't make sense to my that a summary should produce itself via a neural network, unless we are training an auto-encoder.
>
> The intuition behind the recoder is that a good summary should be *useful as input for producing the reference summary*, not necessarily *equal* the reference or "produce itself". This is exactly why we need the recoder. The decoder's output summaries will almost never equal the reference summary, causing a disparity between the sequence prefix that the summarizer sees in training versus test.
>
> This is discussed in the Introduction, so please review that section for further clarification.
>
>
> > I find it odd that ROUGE score is dismissed as a loss to train against ... but then 1 ROUGE point difference is considered a "significant" improvement....
>
> We never dismissed ROUGE, which is why we evaluated using ROUGE in Section 5.1. We said that the recoder can in principle capture more complex notions of quality than ROUGE, so ROUGE *alone* is not enough, and that's why we presented human evaluations in Section 5.2.
>
> I would also say that 1 point is not an insignificant improvement. Most of the reported ROUGE scores for the near state-of-the-art models cited vary within a range of about 2 points, such as 39.47 to 41.69 for ROUGE-1.
>
>
> > making such claims without statistical significance testing is misleading.
>
> I do not see how what we presented is misleading, especially when confidence intervals output by the ROUGE tool was always available in our submitted code and results: https://github.com/iclr2020recoder/code_for_paper . We did not list them in the paper to reduce clutter in the limited space available. This omission is in line with other papers such as Celikyilmaz et al. (2018).
>
> One reason to omit them is that the confidence intervals are generally small (+/- 0.2 points). As an example, here are the results for pgen_cov+recoder (lambda=0.1), which can be seen in the github repository link:
>
> ROUGE-1: 0.4044 with confidence interval (0.4022, 0.4065)
> ROUGE-2: 0.1815 with confidence interval (0.1793, 0.1838)
> ROUGE-3: 0.3690 with confidence interval (0.3668, 0.3712)
>
>
> > Sure it has flaws, if there is something better why not use it for evaluation? Furthermore, claiming that the recoder does a better job requires evidence. Why not train this extra function against human judgements?
>
> We did evaluate using human evaluations as a way to improve on the limitations of the ROUGE heuristic, so I'm not sure what you're asking here. Together these evaluations are evidence of the recoder's improvement.
>
> I'm also not sure what you mean by training "this extra function" against human judgements. It would not make sense to train the recoder using human judgements. You may be thinking of reinforcement learning style algorithms, discussed in Related Work, that train toward a metric.
>
>
> > The approach is in my view a kind of actor-critic approach; the recoder could play that role, and in fact the Bahdanau et al. 2016 paper cited does this. However no comparison is offered, be it theoretical or experimental. Furthermore, the criticism that sequence-level training requires differentiable losses is incorrect; MIXER for example does train against scores such as ROUGE that are not differentiable. Furthermore, the BSO approach by Wiseman and Rush (2016) cited does give a continuous output to optimize for seq2seq that is asked for in the beginning of page 4.
>
> This is all addressed in the Related Work section, which I kindly request that you review, after you have corrected the misunderstandings I noted above. Actor-critic and other reinforcement learning approaches such as REINFORCE are discussed to an extent that we feel adaquate to address what you brought up here. There we discussed how Bahdanau et al. (2016),  Ranzato et al. (2016)'s MIXER, and Wiseman and Rush (2016) are relevant yet different from our work. In Section 5.2 our experimental results are also compared against approaches that use reinforcement learning, such as Chen & Bansal (2018).
>
> You may also have the same misunderstanding as Official Blind Review #3, so please see my response to that review.
>
> <continued in next comment>

---

> > ### Author Response · Authors · 2019-11-05
> > **<continuation of above>**
> >
> > >  how was the length penalty determined to be the graduated curve mentioned in section 5? I would expect to have comparison against other approaches
> >
> > The length penalty function was formed based on intuition and a few adhoc trials. It is a mechanism for controlling length and did not significantly affect quality otherwise, so we did not experiment extensively with it or perform comparisons with other approaches such as those we cited in Section 3.3. One of the strengths of our approach is that we do not require changes to the summarizer model, which the cited alternatives require.
> >
> >
> > > On the human evaluation experiments: the difference between the two models is quite small
> >
> > Our improvements compare favorably against other state-of-the-art work, as discussed in Section 5.2, so I'm not sure why you say this.
> >
> >
> > > the workers were not allowed to say that the models were equally good/bad.
> >
> > We did not allow "equal" ratings because summaries truly equal in quality would be randomly assigned as a win for either side and cancel out in aggregate, so allowing "equal" ratings would only reduce the number of useful ratings by allowing the rater to avoid making a tough decision in close cases.
> >
> >
> > > there is no inter-annotator agreement
> >
> > As mentioned in Section 5.2, we do not need a confident score for any one example, since we are assessing the quality of the algorithm. Ideally we would sample from the space of all (viewer, example) pairs, so limiting each example to one worker allows us to sample a wider diversity of examples using a given number of workers -- we used 300 (viewer, example) pairs for 11490 examples. We could also have limited each worker to 1 example instead of 5 to get a greater diversity of viewers, but we felt that may expose us to a higher fraction of ratings from workers making mistakes on their first and only example.
> >
> > Confident results for each example would only be needed if the ratings were to be used as training data for another algorithm, which is not the case for us.
> >
> >
> > > Showing only the first 400 tokens of the original document would incorrectly disadvantage models selecting content from later in the articles.
> >
> > Both the test model and the comparison baseline (See et al. 2017) were limited to 400 in line with their work, so there is no disadvantage for either.
> >
> >
> > Given the various misunderstandings, we would appreciate it if you could review the paper again after rereading. Thank you.

---

### Official Review · AnonReviewer1 · 2019-10-22
**Official Blind Review #1**

**Rating:** 6

**Review:**

This paper proposed an encoder-decoder-based summarization network as a loss function within a similar encoder-decoder-based summarization framework to demonstrate that the proposed model obtains better automatic and human evaluation scores compared to the baseline model of See et al. (2017) with just traditional loss functions. Overall, the paper is well-written and the presented results, analyses, and comparisons appear to be reasonable. One notable advantage of the proposed model would be to circumvent the approaches that rely on the evaluation metric, ROUGE as a reward component e.g. in a reinforcement learning setting, although with the expense of additional memory and time complexity.

Few comments:

- "A presents either a word ...." --> this sentence is not clear.

- "Embedded representations ... differ somewhat from w_i"--> Please clarify this aspect with more details.

- In Figure 1, the proposed model with recoder seems to be suffering from issues related to redundancy and referential clarity, as it repeats the name "malia" several times. Would you comment on why this is the case?

- It would be great if you could provide more details on the selection criteria/qualifications of the mechanical turk workers. Also, it is not clear why each example was given to only one worker and not to multiple workers. Wouldn't it be ideal to evaluate each example by multiple workers to get a sense of the inter-rater agreement? Please clarify.

**Experience Assessment:**

I have published in this field for several years.

**Review Assessment: Checking Correctness Of Derivations And Theory:**

I assessed the sensibility of the derivations and theory.

**Review Assessment: Checking Correctness Of Experiments:**

I carefully checked the experiments.

**Review Assessment: Thoroughness In Paper Reading:**

I read the paper at least twice and used my best judgement in assessing the paper.

---

> ### Author Response · Authors · 2019-11-05
> **Addressing your comments**
>
> Thanks for your comments.
>
> > "A presents either a word ...." --> this sentence is not clear.
>
> Thanks for catching this. It should be "A token represents either a word ....". Will correct.
>
> > "Embedded representations ... differ somewhat from w_i"--> Please clarify this aspect with more details.
>
> Thanks for this suggestion. We'll add a sentence to clarify that w^R_t is a weighted sum of embedding vectors instead of an individual one like w_i.
>
> > In Figure 1, the proposed model with recoder seems to be suffering from issues related to redundancy and referential clarity, as it repeats the name "malia" several times. Would you comment on why this is the case?
>
> In many cases the coverage mechanism reduces repetition at the phrase level, but usually not down to the level of individual words such as malia. There is no special penalty for redundancy in the standard ml loss, and the same is true for the recoder except from verbosity due to the length loss. Moreover, this dataset's reference summaries, which are usually 3 individual sentences intended as bullet points, may not always resemble a coherent paragraph.
>
> > It would be great if you could provide more details on the selection criteria/qualifications of the mechanical turk workers. Also, it is not clear why each example was given to only one worker and not to multiple workers. Wouldn't it be ideal to evaluate each example by multiple workers to get a sense of the inter-rater agreement? Please clarify.
>
> We required workers to be Masters (a common requirement), which the Mechanical Turk system had identified to be "high performers".
>
> As mentioned in Section 5.2, we do not need a confident score for any one example, since we are assessing the quality of the algorithm. Ideally we would sample from the space of all (viewer, example) pairs, so limiting each example to one worker allows us to sample a wider diversity of examples using a given number of workers -- we used 300 (viewer, example) pairs for 11490 examples. We could also have limited each worker to 1 example instead of 5 to get a greater diversity of viewers, but we felt that may expose us to a higher fraction of ratings from workers making mistakes on their first and only example.
>
> Confident results for each example would only be needed if the ratings were to be used as training data for another algorithm, which is not the case for us.

---

### Official Review · AnonReviewer3 · 2019-10-25
**Official Blind Review #3**

**Rating:** 1

**Review:**

This paper introduces an encoder-decoder as a differentiable loss function for sequential autoregressive generation tasks and more specifically for summarization.
This is done by adding a recorder network that that takes the decoded sequence from the summarizer as input and is trained to output the reference summary.

I see a fundamental issue with this work:

* During inference, authors decode from the probability distribution of the seq2seq model using beam search.
* But for training (original seq2seq + recorder) authors backpropagate the NLL loss (which is fully differentiable) of the recorder on reference summaries through the softmax probabilities of outputs from the seq2seq model.

>> This whole architecture can be seen as a traditional end-to-end seq2seq model with non-linearity and normalization (softmax) in the middle.

Additionally:
>> "backpropagating through the softmax weights during training and using the argmax during inference" falls into a long line of work for propagating non-differential objective functions through continuous relaxations
 of categorical latent variables, more specifically the "straight through"  and "gumbel-softmax" (see refs.)
These methods have proven to be a strong alternative to reinforcement learning to train non-differential objectives and have been implemented quite a lot for sequence generation mainly for SeqGANs and even for text summarization connections to this line of work must be established in this paper.

references
- Estimating or propagating gradients through stochastic ´neurons for conditional computation.
Bengio et al. 2013 arXiv preprint arXiv:1308.3432, 2013.
- CATEGORICAL REPARAMETERIZATION WITH GUMBEL-SOFTMAX
Jang et al. 2018 https://arxiv.org/pdf/1611.01144.pdf
- GANS for Sequences of Discrete Elements with the Gumbel-softmax Distribution
https://arxiv.org/pdf/1810.05739.pdf
- Learning to Ask Questions in Open-domain Conversational Systems with Typed Decoders
https://arxiv.org/pdf/1805.04843.pdf
- MeanSum : A Neural Model for Unsupervised Multi-Document Abstractive Summarization
https://arxiv.org/pdf/1810.05739.pdf






**Experience Assessment:**

I have published in this field for several years.

**Review Assessment: Checking Correctness Of Derivations And Theory:**

I carefully checked the derivations and theory.

**Review Assessment: Checking Correctness Of Experiments:**

I did not assess the experiments.

**Review Assessment: Thoroughness In Paper Reading:**

N/A

---

> ### Author Response · Authors · 2019-11-05
> **There is a core misunderstanding.**
>
> Thanks for your comments. I believe there is a core misunderstanding in how the recoder helps train the summarizer, so let me clarify. The key contribution is the recoder as a *loss function* for the summarizer's beam decoded outputs. What you referred to:
>
> > "backpropagating through the softmax weights during training and using the argmax during inference"
>
> is to help train the summarizer through beam search decoding, which is not an element of our work. The recoder helps train the summarizer only by exposing it to its own output (y1...yt-1), without backpropagation to previous timesteps through beam search decoding. We tried to make this subtle point clear in Section 3.1, where we said "propagating errors to P_t improves the summarizer via exposure to what it would see at test time, even if the summarizer is not 'aware' of the search mechanism and cannot optimize for it".
>
> Propagating through beam search decoding would be accomplished by techniques such as Straight Through Gumbel-Softmax that you cited, or such as Goyal et al. (2017) that we cited. These would be complementary additions to the recoder, as discussed in Section 4.2. We also discussed how our work differs from approaches such as reinforcement learning in Section 4.1. Other models that re-encode their output are discussed in Section 4.3.
>
> The MeanSum work you cited is a related work in a different problem space; thanks for this reference. They also use the idea of re-encoding an output summary, but they differ in that their loss function l_sim is based on cosine distance in the hidden space, in contrast with our recoder loss J^R. Their "reconstruction cycle loss" variant is closer to our work, except they reconstruct the original reviews, which would be analogous in our problem to training the recoder to output the original article instead of the reference summary. We decided against doing this because in general the summary is a lossy representation of the original (mentioned in Section 4.3), an incongruence MeanSum also mentioned as a shortcoming. We will add a citation to MeanSum and Straight Through Gumbel-Softmax and bring up these differences in the paper.
>
> I understand it is difficult to parse out such complexities given a terse presentation, and would appreciate it if you could review the paper again after this very important clarification. Thank you.

---

### Decision · Program_Chairs · 2019-12-19

**Decision:**

Reject

**Comment:**

This paper presents an encoder-decoder based architecture to generate summaries. The real contribution of the paper is to use  a recoder matrix which takes the output from an existing encoder-decoder network and tries to generate the reference summary again. The output here is basically the softmax layer produced by the first encoder-decoder network which then goes through a feed-forward layer before being fed as embeddings into the recoder. So, since there is no discretization, the whole model can be trained jointly. (the original loss of the first encoder-decoder model is used as well anyway).

I agree with the reviewers here, that this whole model can in fact be viewed as a large encoder-decoder model, its not really clear where the improvements come from. Can you just increase the number of parameters of the original encoder-decoder model and see if it performs as good as the encoder-decoder + recoder? The paper also does not achieve SOTA on the task as there are other RL based papers which have been shown to perform better, so the choice of the recorder model is also not empirically justified. I recommend rejection of the paper in its current form.